# Religion, Populism and Politics: The Notion of Religion in Election Manifestos of Populist and Nationalist Parties in Germany and The Netherlands

**Leon van den Broeke [1,2,\*] and Katharina Kunter [3,\*]**

1 Faculty of Religion and Theology, Vrije Universiteit Amsterdam, 1081 HV Amsterdam, The Netherlands
2 Department of the Centre for Church and Mission in the West, Theological University, 8261 GS Kampen, The Netherlands
3 Faculty of Theology, University of Helsinki, 00014 Helsinki, Finland
\* Correspondence: c.vanden.broeke@vu.nl (L.v.d.B.); katharina.kunter@helsinki.fi (K.K.)

**Abstract:** This article is about the way that the notion of religion is understood and used in election manifestos of populist and nationalist right-wing political parties in Germany and the Netherlands between 2002 and 2021. In order to pursue such enquiry, a discourse on the nature of manifestos of political parties in general and election manifestos specifically is required. Election manifestos are important socio-scientific and historical sources. The central question that this article poses is how the notion of religion is included in the election manifestos of three Dutch (LPF, PVV, and FvD) and one German (AfD) populist and nationalist parties, and what this inclusion reveals about the connection between religion and populist parties. Religious keywords in the election manifestos of said political parties are researched and discussed. It leads to the conclusion that the notion of religion is not central to these political parties, unless it is framed as a stand against Islam. Therefore, these parties defend the Jewish-Christian-humanistic nature of the country encompassing the separation of 'church' or faith community and state, the care for the historical and cultural heritage of church buildings, and the subordination of the freedom of religion to the freedom of expression. The election manifestos also reveal that Buddhism and Hinduism are absent in the discourses of these political parties.

**Keywords:** election manifestos; religion; populist and nationalist political parties; Germany; The Netherlands

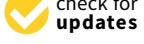



## 1. Introduction

This article investigates the way in which religion is understood and used in election manifestos of populist and nationalist right-wing political parties in Germany and the Netherlands between 2002 and 2021, see Table 1. The impetus for this explorative research is the rise of nationalist and populist voices around the globe and more specifically, in Dutch and German politics as well as the current academic discussion on the entanglement of religion and the rise of populism (Nillson and Shterin 2019). In both countries, Christian-democratic political parties have exerted great influence in the political arena (again, in the Netherlands, since 2002, after the two purple coalition governments; in Germany, since 2005). Moreover, both countries have generally worked well together as partners in the European Union. In both countries, populism and nationalism are not new phenomena, but today they form their own societal, socio-economic and (geo-)political context. They are reactions to the (geo-)political and socio-economic developments that have taken place since the early 20th century. They oppose national and European governments, societal aspects such as (in)security and terrorism following the attacks on 9/11, the Islamization of society and the European migration crisis of 2014/15 and beyond.

**Table 1.** Overview of Dutch and German party manifestos 2002–2021.

| Election Manifesto | Year | Political Party | Country |
|---|---|---|---|
| Business with a Heart | 2002 | LPF | The Netherlands |
| Politics with Passion | 2003 | LPF | The Netherlands |
| This is not the country I would like to leave behind for my children | 2006 | Lijst 5 Fortuyn | The Netherlands |
| Election Manifesto | 2006 | PVV | The Netherlands |
| The Agenda of Hope and Optimism: A Time to Choose | 2010 | PVV | The Netherlands |
| Their Brussels, our Netherlands | 2012 | PVV | The Netherlands |
| Draft Election Program Party for Freedom | 2017 | PVV | The Netherlands |
| It is about you | 2021 | PVV | The Netherlands |
| Draft Election Manifesto | 2017 | FvD | The Netherlands |
| Vote the Netherlands Back | 2021 | FvD | The Netherlands |
| Bundesparteiprogramm | 2002 | Die Republikaner | Germany |
| Programm für Deutschland | 2017 | Alternative für Deutschland (AfD) | Germany |
| Europawahlprogramm | 2019 | AfD | Germany |

Koen Vossen (2012) distinguishes between three populist eras in the Netherlands: 1916 to 1918 (The First World War), 1963 to 1967 (the 1960s) and finally, the period beginning in 2001/2002 (9/11 and the murder of Pim Fortuyn). On 20 August 2001 Fortuyn publicly declared that he wanted to become involved in Dutch politics. It was three weeks before the 9/11 attacks on the United States of America. Eight months later the promising, popular and controversial politician Fortuyn was assassinated.

In Germany, Dutch politics and the murder of Fortuyn garnered much attention but was considered to be a particular Dutch phenomenon rather than expressing a general development (see, e.g., headline Der SPIEGEL 5 May 2002: "Der erste politische Mord der Niederlande" ("first political murder of the Netherlands"): https://www.spiegel.de/politik/ausland/fortuyn-erschossen-der-erste-politische-mord-der-niederlande-a-195099.html, accessed on 30 January 2021). Such assessment was incorrect and this became clear in the time that followed. Nationalism and populism are on the march not only in the Netherlands, but also in Germany and in the rest of the world.

Although the Netherlands and Germany are political neighbors with many cultural points of convergence and similar European political interests, the emergence of right-wing populist parties and movements in the two countries shows the differences between them. In 2011, Wielenga and Hartleb (2011, p. 8) stated that the most noticeable difference between the Netherlands and Germany is that right-wing populism had seemly gained a secure place within the Dutch political party spectrum since the meteoric rise of Pim Fortuyn in 2002 (Fortuyn 2002), and this is not the case in Germany. The latter's history of National Socialism from 1933 to 1945, the reunification of Germany in 1990 and a strong Federalism had delayed the development of right-wing populist parties. The different voting regulations of the Netherlands and Germany with regard to the rights of smaller parties also played a role. Only with the foundation of the "Alternative für Deutschland" (AfD) in 2013 and its entry into the German Parliament, one can speak of a nationally successful populist right-wing party. Even if the history and political structures of the two countries are very different, it is worthwhile taking a comparative look at the position of religion in the party programs. This comparison, which is based on a selected source available in both countries (election manifestos of populist parties), also appears instructive because populism researchers like Roger Brubaker distinguish a distinctive

difference between the two countries: while in the Netherlands he sees the new populist type of civilization rhetoric (Brubaker 2017, p. 1196), Germany falls out of this framework (Brubaker 2017, p. 1193).

The main theme of this article is to show how the notion of religion appears in the election manifestos of populist and nationalist parties in Germany and the Netherlands, and to answer what such appearance reveals about the connection between religion and populist parties.

What is subsumed under religion? Which religious arguments arise in the party manifestos? Which relationship between Christianity, Judaism and Islam is shown to the outside world? To what extent does religion have a cultural component, and if so, is it used strategically? And how clearly do the manifestos openly express their anti-Islamic stance? With these questions, too, one should keep an eye on the extent to which religion, especially Christianity, is used in the Party Manifestos in the sense of "othering" and to what extent a new "identitarian Christianism with secular and liberal rhetoric" (Brubaker 2017, p. 1194) is constructed.

This article answers these questions in the following sections. Part two focuses on party manifestos as historical and political sources in general. In section three the party manifestos of three Dutch populist and nationalist parties in the period between 2002 and 2021 are examined regarding the notion of religion. Part four pursues the same question regarding German politics and the final part of the article answers the questions posed and presents the conclusions of the research.

## 2. Election Manifestos

This study is based on the election manifestos of populist right-wing parties in Germany and the Netherlands. As already noted in the introduction, the historical preconditions for the emergence of populist parties are different in the two countries and the election manifestos therefore cannot be compared and contextualized on the same empirical basis. For example, right-wing populist parties emerged and successfully gained entry into parliament earlier in the Netherlands than in Germany as well as gaining wider national and international media attention. The elections did not take place at the same time and the party programs differed in terms of publication dates and cultural origin.

Nevertheless, it can be said for both countries that election manifestos are an important socio-scientific and historical source. They are 'coherent statements of policy ( . . . ) [and] they indicate which populations are most significant to the party's desired electoral outcome' (Harmel 2018, p. 230). These functions represent the outward face of manifestos, seeking to appeal to voters (cf. Harmel et al. 2018, pp. 278–88).

There are further two functions that can be distinguished: an internal focus, wanting 'to smooth over internal differences' and an agenda-setting focus, attempting 'to help shape the broader issue agenda over which an election is fought' (Harmel 2018, p. 231). Considering these three foci, party manifestos reflect that which a party deems important, give an insight into internal party debates and point to the issues which party supporters consider to be important electoral issues. In research, different methodical approaches are used to understand and analyze election manifestos: In the political and social sciences, quantitative and empirical election manifestos research has prevailed (Volkens et al.; Merz and Regel 2013, pp. 146–68). Here two major directions are dominant: Comparative Manifesto Project (Franzmann and Kaiser 2006) and the word score analysis (Bräuninger et al. 2012). In historical sciences, where party programs function as historical sources, hermeneutic and semantic approaches are used in addition to the social science methods mentioned, such as discourse analysis. The latter is also used in the social sciences. In honoring the interdisciplinary character of the article, it makes use of both empirical and cultural-historical methods. By combining these methods, the discourse analysis explores the parties' views on religion embedded within a cultural framework, revealing fundamental ideas about what they consider to be the ideal and desirable character of today's Europe, nation and society.

This raises the issue of the relationship between election promises made in the manifestos and the political reality arising when political parties become part of a new government. The issue of breaking promises should not be ignored and Naurin (2011, p. 23) concludes that parties 'risk not only their voters but also their own party members' when breaking promises made in election manifestos. Moreover, Naurin (2011, p. 3) states that:

> "The levels of congruence between promises and actions differ among the different systems that are studied, but there is a clear common denominator in the conclusions that are drawn: promises given in election manifestos are taken seriously by the governing political parties."

Still, Erin Naurin (2011, p. 20) points out that manifestos are important in relation to election promises and the perception of voters:

> "Richard Rose, who has published one of the British studies on election promises, even describes election manifestos as more important to the party itself than to its voters: 'the drafting of a manifesto is first of all a search for consensus within each of the parties. The resulting document is not so much a statement of what the voters want as it is a proclamation of what a party's leadership agrees to want' ( . . . ). The process of deciding policy programmes in this way is in itself a way to tie the more or less divided party together. If a party succeeds in writing a coherent and credible election manifesto, it can be assumed to be strong in the respect that it will hold together and work towards a collective goal. The writing of the election manifesto therefore provides important information about the party."

Election manifestos are ideal-typical. They contain the 'ideal' society in the vision of the specific political party. However, it is the society of a broken world and an imperfect society which contains much diversity and conflicting political, social, economic, religious and cultural standpoints and world views. Manifestos are not always applicable to the political arena and to public life. Still, they provide a vision of an almost perfect society.

Nonetheless, their value should not be underestimated: "They can be understood as the political parties' most important priorities" (Naurin 2011, p. 19) and they can be considered 'as an important tool for the parties in trying to understand citizens' priorities" (Naurin 2011, p. 18).

A last remark on party manifestos is fourfold. In order to be compared to manifestos of other parties they should be an 'election program' or 'electoral manifesto' (Merz and Regel 2013, p. 149). Two, the manifesto should reflect the position of the whole party. Three, the manifesto needs to be related to elections and finally, the document should 'reflect the party's programmatic profile' (Merz and Regel 2013, p. 150).

## 3. Election Manifestos of Dutch Populist and Nationalist Parties 2002–2021

This section focuses on the notion of religion in the election manifestos of Dutch populist and nationalist parties between 2002 and 2021. However, within a Dutch context election manifestos predate the 20th century. When Abraham Kuyper, an erstwhile minister in the Netherlands Reformed Church, a journalist, a (political) minister and Prime Minister, established the first political party in the Netherlands in 1879 namely the Anti-Revolutionary Party (one of the predecessors of the CDA, the Christian Democratic Appeal), he wrote the manifesto *Ons Program* (Our Program) (Kuyper 1879). This manifesto consisted of 1307 pages. In the first sentence of the Introduction, Kuyper (1879, p. 7) stated that a manifesto can be threefold in nature: it is a party manifesto (detailing the principles of a party), it is an election manifesto and/or a manifesto of government. As no general election was imminent, Kuyper (1879, p. 11) considered *Ons Program* a party manifesto, detailing the principles of the party.

From 1917 onwards and prior to the establishment of the populist political parties in the early 2000s, there have been political parties in the Netherlands which can be identified as populist parties: the Rural Association (1917), the Farmers' Party (1958),

Centre Democrats (1984) and Livable Netherlands (1999) ([Populisme n.d.](#)). As we have already mentioned, we focus on the populist and nationalist political parties established between 2002 and 2021. Therefore, the election manifestos of the following Dutch political parties have our interest: the LPF (*Lijst Pim Fortuyn*), PVV (Party for Freedom, Geert Wilders), and FvD (Forum for Democracy, Thierry Baudet). After introducing these parties, our focus will be on their party manifestos with regard to religion.

### 3.1. LPF: Lijst Pim Fortuyn

3.1.1. The Party

It was a unique occurrence in the history of Dutch parliamentary democracy that a new political party, the LPF, gained 26 seats (of altogether 150 seats, amounting to 17.33% of the votes) in the Second Chamber, and became part of the government. Together with the Christian Democratic Appeal (*Christen-Democratisch Appèl*, CDA) and the People's Party for Freedom and Democracy (*Volkspartij voor Vrijheid en Democratie*, VVD), the LPF governed the Netherlands between 22 July 2002 and 27 May 2003. However, the political founder and leader of the LPF, Pim Fortuyn, would never have been part of this coalition government. Nine days before the elections of 15 May 2002, the political leader of the new political party, the LPF, was murdered. He had aimed to become the next Prime Minister ([LPF en Tweedekamerverkiezingen 2002](#)).

3.1.2. Business with a Heart 2002

The election manifesto *Zakelijk met een hart* (Business with a Heart) attacked eight years of two purple coalition governments. This manifesto consisted of nine pages and fourteen chapters or paragraphs. There was no chapter specifically on religion and in general, the election manifesto contained few references to religion. This was also the case regarding the Jewish-Christian-humanistic developments in Europe. The LPF believed that large groups in society had little or no social-cultural "skills" as they came from countries lacking Jewish-Christian-humanistic developments. As Fortuyn was very critical of Islam, Muslims, migrants and immigration, it is revealing that the election manifesto only included minor references to such topics.

3.1.3. Politics is Passion 2003

Only a year after the previous election manifesto had been written, a new manifesto was required due to the resignation of the government [Balkenende 1]. The new manifesto of the LPF of 2003 was roughly 3 and a half times longer than the previous manifesto ([LPF en Tweedekamerverkiezingen 2003](#)). It was given the title *Politiek is passie* (Politics is Passion). It was based both on the previous manifesto—*Zakelijk met een hart*—and Fortuyn's book *De puinhopen van acht jaar Paars* (*The Mess of Eight Purple Years*). The word 'Paars' referred to the coalitions of the blue liberal party VVD (People's Party for Freedom and Democracy) and the red social democratic parties PvdA (Labour) and D66 (Democrats 66) in the coalition government (Wim) Kok I (1994–1998) and Kok II (1998–2002). These were the first coalition governments since 1918 that did not include a Christian democratic party, like the CDA. Fortuyn heavily criticized these purple coalition governments. With his new political party and his aim of becoming the next Prime Minister, he wanted to renew both the political arena and Dutch society.

After the death of Fortuyn, divisions within the party on such matters as his political heritage and leadership appeared. Due to inner-party conflicts, the LPF lost 18 seats (from 26 seats down to 8 seats) in the Second Chamber, receiving 5.7% of the votes compared to the 17% that it had won in 2002.

The manifesto included fifteen chapters or paragraphs, but there was no specific chapter on religion. In general, the election manifesto had few references to religion. It contained five references to Islam, one to religion in general and two to the relationship between state and church. The introduction of the manifesto *Politiek met passie* referred to the convictions of Fortuyn who had warned about the collision of two dominant cultures:

modernity and Islam. Therefore, Fortuyn stressed the importance of the separation of church and state, but only a few times (Politiek is Passie 2003).

Where integration was concerned, a societal debate about the role of Islam in Dutch society was required as the Islamic culture represented the opposite of the values and norms of modern Western society. The manifesto made a plea for a spirited dialogue to take place between the two cultures rather than ignoring one another. Such dialogue should lead to mutual understanding. Not least for the sake of living together side by side in society the program strove for furthering the emancipation of Islamic women according to the Declaration of the Universal Rights of People.

In the paragraph on Europe, Defense and Development Aid, the manifesto made quite clear that the most important field concerning the clash between Western society and the Islamic culture was probably that of the separation of church and state.

Additionally, when it came to the topic of Administration and Bureaucracy, the LPF wanted to initiate a societal debate where tension between the respective articles of the constitution: freedom of expression, freedom of religion, freedom of education and anti-discrimination were visible.

Regarding the naturalization process, the manifesto wanted new citizens to actively participate in Dutch society. This included the acceptance of Dutch regulations and norms, for example the separation of church and state.

### 3.1.4. This Is Not the Country I Would Like to Leave behind for My Children 2006

In 2006 the name of the LPF was changed into *Lijst 5 Fortuyn*. The election manifesto of 2006 was entitled: Dit is niet het land wat ik voor mijn kinderen wil achterlaten (This is not the country I wish for my children) (Dit is niet het land wat ik voor mijn kinderen wil achterlaten 2006). The *Lijst 5 Fortuyn* received 0.2% of all the votes in the Netherlands. A year later the party ceased to exist.

The election manifesto 'Dit is niet het land wat ik voor mijn kinderen wil achterlaten' (This is not the country I would like to leave behind for my children) was 23 pages long and focused on five main themes: participation, revaluation of professionals, senior citizens, immigration/integration and EU, and young people.

In comparison to the previous manifesto, it is revealing that the election manifesto of 2006 only refers one time to Islam. When it comes to the topic of immigration and integration the *Lijst Vijf Fortuyn* wants to put a stop to the building of new mosques and Islamic schools.

The manifesto refers two times to Fortuyn who, with a background in sociology, advocated the separation of church and state. According to both him and the manifesto such separation is not negotiable. This, however, does not mean that religion has not been important. Despite the fact that fewer people belong to a religious community, religion is still considered important to people, but more as a private matter; this stressed the requirement for the separation of church and state. Three other aspects were also considered to be unnegotiable: freedom of expression, tolerance and non-discrimination.

Fortuyn considered integration as being a combination of religious and political aspects. Therefore, he warned against Islam and immigration as he wanted to protect the Dutch nation because of its values on the separation of church and state, the development of parliamentary democracy, the equality of men and women, of homosexuals and heterosexuals, the free market, freedom of expression, individual responsibility and a sense of community.

### *3.2. PVV*
### 3.2.1. The Party

The PVV was established in 2005 or 2006. Its political leader is Geert Wilders. It gained 9 seats (5.9%) in 2006; 24 seats (15.5%) in 2010; 15 seats (10%) in 2012 and in the elections of 2017 20 seats (13.1%).

### 3.2.2. Election Manifesto 2006

The election manifesto of the PVV of 2006 included nine chapters or paragraphs. The manifesto did not have a chapter on religion and contained few references to the notion of religion. In paragraph IV on Immigration Stop/Integration of the *Verkiezingspamflet* [Election Manifesto] 2006 one, of three main aspects, was a new article 1 of the Constitution (Verkiezingspamflet 2006). This new article should make clear that the dominant culture in the Netherlands remained the Christian-Jewish-humanistic culture. Additionally, there was a ban of five years on building new mosques and Islamic schools. Radical mosques should be closed, and radical imams should be evicted. There had to be a prohibition of foreign financing of and influence on mosques in the Netherlands. Foreign imams should be prohibited to preach in Dutch mosques. Moreover, it was stipulated that the Dutch language should be spoken in these mosques.

In this way Wilders and his PVV wanted to protect the Jewish-Christian-humanistic nature of the Netherlands. He emphasized the need for a small administration and a main focus on the civil society with societal organizations, families and churches. Nothing was said about religion in general in the manifesto except for stressing the Christian-Jewish-humanistic nature of the Netherlands. It is revealing though that the PVV used a different order by putting 'Christian' before 'Jewish' in the characterization of the country.

### 3.2.3. The Agenda of Hope and Optimism: A Time to Choose 2010

The election manifesto *De agenda van hoop en optimisme: Een tijd om te kiezen* [The Agenda of Hope and Optimism: A Time to Choose] from 2010 was a 60 pages long manifesto, divided into thirteen chapters. Each of these chapters began with the word 'choosing'. There is no specific chapter on religion. The manifesto emphasized the Christian-Jewish-humanistic values and nature of the Dutch society (De agenda van hoop en optimisme, een tijd om te kiezen: PVV 2010–2015 (parlement.com), accessed on 14 January 2021). According to the program, these values made the Netherlands a success. Therefore, Christian, Jewish and public schools could exist side by side, but Islamic schools should be closed.

This election manifesto referred to Islam 42 times and in a negative way. For example: "Islam does not bring cultural enrichment, but sharia-fatalism, Jihad terrorism and hate against homosexuals and Jews. Everywhere in Europe we face the same problems with Islam." (De agenda van hoop en optimisme 2010, p. 6) The next paragraph was begun with "those who want to participate: welcome." Furthermore, it firmly stated that a moderate Islam does not exist. According to the PVV, Islam is built on two pillars: the literal interpretation of the Qur'an and the perfectness of the prophet Mohammed. Moreover, Islam divides people in two categories: Muslims and *kaffirs* (non-Muslims). The Jihad, the holy war, is an assignment, a holy duty for Muslims. They have to fight against our rule of law, democracy, the equality of men and women, and the Jewish people.

The manifesto refers one time to the separation of church and state as a result of the dominant Jewish-Christian-humanistic culture. The inclusion of Jewish culture is not only a historical consideration, because Jews have been living in the Netherlands for centuries, partaking in Dutch society. It is also motivated by the pro-Israel attitude of the PVV regarding the geo-political context. The manifesto states that Israel is a tremendous success. and fights for the Netherlands. In the eyes of the PVV, Israel is the central frontier in the defense of the West, when Jerusalem falls, Athens and Rome will be next—the entire West-European culture. Israel is prominently present in this manifesto. It fits in the context of those days: the possibility of a third Intifada and/or the war in Gaza.

The PVV suggested the following solutions regarding Islam:

- as Islam is a political ideology, it is not eligible to obtain the rights of freedom of religion;
- no more (new) mosques;
- the closing of all Islamic schools;
- the closing of mosques in which violence is preached;
- no Islamic diversity between the male and female sexes;

- no financial support for Islamic media;
- reduction of the budget of multicultural Netherlands;
- no scarves to be worn in health care, education, city hall, civil authorities, or any subsidized organizations.

The PVV was angry that religious broadcast companies such as the Roman Catholic KRO did not make one Roman Catholic program and that the inter-religious IKON abused time on dismissing the PVV as the National-Socialistic Movement (NSB) of the 1930s.

3.2.4. Their Brussels, Our Netherlands 2012

The 53 pages long election manifesto *Hún Brussel, óns Nederland* [Their Brussels, our Netherlands] 2012 contained ten chapters. Apart from the first chapter (Their Brussels), each of the next chapters began with the word 'Our'. This election manifesto did not contain a chapter on religion either, but it did refer to the notion of religion. The PVV favored the Jewish State as it was a beacon of hope, progress and western civilization. The process of bashing Israel by allied Islamic forces and left-wing political parties at all levels should be stopped. Therefore, the PVV did not want the anti-Israel hate-industry to be subsidized any longer and political support should be given to the building of Jewish villages in Judea and Samaria (Israel). Finally, support—in the sense of human rights— should be given to threatened minorities such as the Christians in Egypt (the Copts) and Armenia.

The manifesto did not specify wherein the Christian and/or Jewish essence of the Netherlands consisted, nor did it include anything about church(es). The election manifesto was more focused on Islam. The PVV states that Islam is not a religion, but a totalitarian political ideology with little religious taint (Hún Brussel, óns Nederland 2012). In this manifesto the headscarf or the burqa represents an issue and the PVV opposed its use by female Muslims in the public domain. Moreover, the PVV was against mosques in the Netherlands.

3.2.5. Draft Election Program Party for Freedom 2017

The *CONCEPT-VERKIEZINGSPROGRAMMA PVV* [Draft Election Program Party for Freedom] 2017–2021 states: *Nederland weer van ons*! [The Netherlands again of us] (Nederland weer van ons! 2017). This draft only contains one page. It was not followed by a final manifesto, but by an explanation of four pages by Geert Wilders. The draft begins by stating that millions of Dutch people have had enough of the Islamization of the Netherlands, enough of mass immigration and asylum, terrorism, violence and are feeling unsafe. The PVV wants to grant the money to the ordinary Dutch people, and not finance the whole world and those who do not want to have it in the Netherlands. The PVV wants to reach its goals by employing a set of "tools":

- to de-Islamize the Netherlands;
- the Netherlands should again be independent;
- direct democracy;
- to abolish own risk health care;
- to reduce rent money;
- to reduce the pension age;
- no more funds to be spend on development aid, windmills, art, innovation, broadcasting, etc.;
- to rectify the deficit in the budget for home care, care for senior citizens ( . . . );
- higher spending for the defense and the Police;
- lower income tax;
- reduction of car tax.

The process of the de-Islamization of the Netherlands was the primary tool. For the PVV this meant (Nederland weer van ons! 2017):

- zero refugees and no immigrants from Islamic countries, closing of borders;

- the withdrawal of all residence permits given to asylum seekers for a fixed term; the closing down of refugee shelters;
- no Islamic scarves to be worn at public functions;
- the prohibition of other Islamic forms of expression which contradict public order;
- radical Muslims to be put in prison prior to an actual offense;
- de-naturalize criminals who hold double nationality and evict them;
- no return to the Netherlands possible for those who fought in the Syrian War;
- all mosques and Islamic schools to be closed, prohibition of the Qur'an.

This program did not include anything on religion in general—except for Islam—and church(es).

Furthermore, it is revealing that a political party that bears the word 'Freedom' (in Dutch: *Vrijheid*) in its name has little to say about freedom. The word 'Freedom' is not merely part of the name of this party, but it should reveal something about the identity of said party.

### 3.2.6. It Is about You 2021–2025

The 52 pages long election manifesto *Het gaat om u* (It is about you) contains twelve chapters. All chapters begin with the word 'Your'. The PVV wants to address its supporters and voters in a direct way. No chapter focuses on religion, although the PVV states that it is of the 'utmost importance to lay down in the constitution that our Jewish-Christian and humanistic roots form the dominant and leading culture in the Netherlands' (Het gaat om u 2021, p. 10).

### 3.3. *Forum voor Democratie*
### 3.3.1. The Party

The relatively new political party *Forum voor Democratie* (Forum for Democracy) was founded in 2016. It participated for the first time in the elections for the Second Chamber of the Dutch Parliament and gained two seats, 1.8% of the votes (Forum voor Democratie (FvD) (n.d.)). It is mainly focused on direct democracy and national sovereignty.

### 3.3.2. Draft Election Manifesto 2017–2021

The draft election manifesto of the FvD 2017–2021 does not have a title, except for *Concept Verkiezings Programma* [Draft Election Manifesto] 2017–2021 (Verkiezingsprogramma FvD 2017). This draft consists of 30 pages and 26 chapters. There is no specific chapter on religion. Chapter 16 concerns Geostrategy. It refers to the change of regimes in the Middle East that has happened with the help of Western countries, leading to a more serious security situation. Christians and also other religious, cultural and ethnic minorities and homosexuals suffer due to the Arabic Spring. This is reflected within an European context. Among other things the FvD wants support for the position of Christians in the Middle East. The FvD has a different approach for adherents of religions in the Netherlands: "Adherents of religions need to respect the plurality of all world views, also of competing religions, of humanism and atheism, of critics, scientists and cartoonists. The freedom of expression has priority over the freedom of religion." Moreover, the FvD opposes foreign financing of domestic religious schools and other religious institutions. Religious schools are required to teach other religions and worldviews other than their own. This includes remembering the Holocaust.

### 3.3.3. Vote the Netherlands Back 2021

The election manifesto of the FvD 2021 contains 104 pages. The title is *Stem Nederland terug* (Vote the Netherlands Back). It consists of seven chapters. There is no specific chapter on religion. An anti-Islam perspective is hardly present in the manifesto, especially in comparison to the manifesto of the PVV. Headscarf/burqa/niqab is mentioned only twice, and Jihad thrice. Moreover, the manifesto does not refer to Allah, Muhammad, mosque or

the Qur'an. However, the manifesto does mention 'Islam' five times, which is five times more than Christianity, and Judaism is absent although 'Israel' (7*) and 'Jerusalem' (1*) are important enough to the FvD to be included in the election manifesto. At the same time, the manifesto emphasizes twice the Jewish-Christian tradition and values of the Netherlands, and also the classic-humanistic world(view).

The Bill Protection Dutch Values is back on the agenda of the FvD. This bill stipulates that every institution needs to subscribe to five fundamental principles. All five principles are important to faith communities and their adherents:

1.　　if religious regulations conflict with Dutch law, the latter takes priority;
2.　　everyone has the right to believe whatever he or she wants. This includes the right not to believe (any longer);
3.　　everyone has the right to criticize religious ideas in the broadest sense of the word;
4.　　all people are fundamentally equal, despite gender, ethnicity and/or sexual orientation;
5.　　everyone has the freedom to choose his or her partner. This means that child marriages and forced marriages are unacceptable (Stem Nederland terug 2021, p. 18).

The FvD values the existence and preservation of historic church buildings. It fits into the narrative of the protection of Dutch historic heritage.

## 4. German Populist and Nationalist Parties

As briefly explained in the introduction, the situation in Germany differs from that of the Netherlands in as far as a broad, politically successful right-wing populist party only emerged with the foundation of the Alternative for Germany (AfD) in 2013. Nevertheless, the Western part of Germany, the Federal Republic of Germany (FRG), has seen precursors such as the nationalist, right-wing extremist "Nationaldemokratische Partei Deutschlands "(National Democratic Party of Germany, NPD), which was founded in the tradition of German National Socialists in 1964. Another right-wing extremist movement with populist features was the "Deutsche Volksunion" (German People's Union", DVU) founded in 1971. The DVU changed its organizational form from being an association (Verein) into a party in 1987 and was subsequently successful in nine state elections, but then merged with the NPD in 2010. Four years earlier, a right-wing Conservative split-off from the Christian Conservative regional Party of the Federal State of Bavaria, the "Christlich Soziale Union" (Christian Social Union, CSU) formed the party "Die Republikaner" ("The Republicans", REP).

They were, as the DVU, unsuccessful in federal elections, but successful in regional state elections, especially in Southern Germany in the 1990s and in the European elections in 1989. In terms of content, they increasingly orientated themselves towards the DVU and the NPD and represented nationalist-ethnic positions in their party programs (Butterwege and Meier 1997, p. 62). Its original roots in Christian (especially Catholic) social ethic is then also expressed in the fact that the Christian religion is still addressed in their federal party manifesto of 2002, divided into four sections: (1) In the area of school education, priority should be given to "Christian-Western" culture (Bundesparteiprogramm 2002, p. 19), (2) in the section "Foreigners " where they state that if foreigners are unwilling to integrate, especially Islamist fundamentalists, they should have their residence permit withdrawn (Bundesparteiprogramm 2002, p. 20), (3) in the "Animal protection" section, they make, with regard to the German Constitution of Article 4, Paragraph 2a, a strong plea for animal protection—that is here related to the ritual slaughtering of animals without stunning as it is used in Islam and Judaism (Bundesparteiprogramm 2002, p. 49). Finally, a section about religion and the churches, which concludes the party manifesto (Bundesparteiprogramm 2002, p. 50). This section stresses the special role of religion for a "vigorous" ("lebenskräftiges Volk") people and emphasizes that for Germany this can only be Christianity. Judaism and Islam thus do not belong in Germany, whereby in the context of the party manifesto of the DVU, it is more about Islam than Judaism. A multicultural society is expressly rejected. Germany's

Christian character, which is interpreted as an expression of Western culture and regarded as the foundation of state and society, of family, democracy and social justice, should be defended. The election manifesto is critical of churches as the institutional formation of Christianity in Germany if they are not focused on preaching the Gospel, but on "preaching" social-ethical issues. This is interpreted as an "adaptation to the Zeitgeist" and as a "left" orientated church. In its federal party manifesto the "Republikaner" call the churches to return to their mission and plead for the separation of church and state. In this context they also speak against a state organized church tax (which does not exist in the Netherlands) and against church asylum ("Kirchenasyl"). Church asylum is the last, legitimate attempt (*ultima ratio*) of a community to help refugees by granting them temporary protection in order to work towards a renewed, careful review of their situation and this was first applied in West Germany in 1983.

While the federal party manifesto openly opposed Turkey's accession to the EU, it did not justify its opposition on religious grounds; the European Party manifesto now expressly names Islam as a reason. Islam is equated with Islamization as the manifesto states that if Turkey would join the EU, Europe would become inexorably Islamized. This explains why the Republikaner demand that Turkey's accession to the EU must be stopped in order to preserve Europe's identity as a Christian Occident (Europaprogramm 1989). Several aspects follow the federal party manifesto, but overall Islam is more clearly emphasized as a religious and cultural threat. For example, the term "Muslims" appears for the first time. On the other hand, fewer references are made to a Christian dominant culture.

At the same time as "Die Republikaner" formulated their ideas in their federal party manifesto, another, newly founded populist party came into the public eye: the "Partei Rechtsstaatlicher Offensive" (Rule of Law Party) of the Hamburger judge Ronald Schill. Founded in 2000, it was part of the Hamburg government from October 2001 to March 2004 but failed to move into the Bundestag and was dissolved in 2017. In public, the party focused on inner security, i.e., law and order and spoke out in favor of a restrictive "foreigners' policy" and rejected the current asylum law. It refused to allow Turkey to join the EU. However, at this point in time—before September 11, 2001—the discussion of these topics did not take place under the heading of "religion" (Schmitz Michael 2002). This marked a clear difference to "Die Republikaner" and programmatically showed a more religiously distant urban character.

While the above-mentioned right-wing populist and right-wing extremist parties were only successful regionally in federal state elections, the "Alternative für Deutschland" ("Alternative for Germany", AfD) founded in 2013, was the first right-wing populist and partly right-wing extremist party that was successful nationwide in the 2017 federal election to the "Bundestag" and in the 2019 European Union elections. It has since then been represented in the German Parliament. Religion appears as a central factor in their party manifestos of 2017 (Federal) and 2019 (Europe); in both with specific sections on "Religion". However, the headings of these sections and the word score analysis clearly shows that the keyword "religion" is fundamentally only about Islam. This appears 12 times in both election programs, followed by the term "Moschee/vereinigungen" ("mosque/associations"), which is used 5 or 3 times in both programs. All other terms appear only once, whereby terms associated with Islam such as "head scarf", "burqa", "niiqab" (only once) have no Christian-Jewish equivalents.

Section 6 of the federal party manifesto of 2017 provides the guiding principle (Programm für Deutschland 2017, Section 6, pp. 33–34): "Islam is in conflict with the "freiheitlich-demokratischen Grundordnung" ("free and democratic basic order"). Right at the beginning it is stated that Islam, with its five million Muslims in Germany, does not belong to Germany and represents a great danger to the state, society and German values. However, it is acknowledged that many Muslims are integrated in Germany and live in accordance with the constitution. However, the AfD wants to prevent "Muslims becoming radicalized to violent Salafism and terror" and "isolated Islamic parallel societies", in which Islamic law, the "Sharia", applies. Therefore, only the civil marriage (under the auspices

of the state) should be binding; religious weddings and pre-weddings, compulsory ceremonies conducted abroad, child marriages or polygamous marriages should not be recognized in Germany. The AfD recognizes the freedom of belief, conscience and confession anchored in Article 140 of the German Constitution. However, they assume that Islamic organizations in Germany do not meet the legal requirements of the free state church law and should therefore not be granted the status of a corporation under public law ("Körperschaft Öffentlichen Rechts"). In cases where religious traditions and commandments come into conflict with state law, the practice of religion should be restricted. In this context, criticism of Islam and religious satire are also permitted; it should not be defamed as "Islamophobia" and "racism".

With regard to the practical exercise of religion, the AfD rejects the minaret and the muezzin as Islamic symbols of power and "religious imperialism"; in their opinion, they contradict a tolerant coexistence of religions, as practiced by Christian churches, Jewish communities and other religious communities. Sermons in mosques are to be given in German, sermons that are unconstitutional must be prohibited from being preached and imams deported from Germany. The construction, operation and financing of mosques supported by anti-constitutional associations or Islamic states should be prohibited. A "culture war" is being waged here from outside, especially due to the dependence of Turkey's state "Office for Religious Affairs" (Diyanet). In order to prevent this, all Islamic theological chairs at German universities should be abolished, instead denominationally neutral Islamic studies should be established at universities. Finally, the AfD advocates a general ban on full veiling in the public area and in the civil service. Burqa and niqab are obstacles to a successful integration due to their religious and political subordination.

Following this specific section on Islam, references to Islam are made several times in the following sections of the party manifesto. For example, it is opposed to denominational Islamic classes in German schools, and religiously based "special rights for Muslims" in schools are rejected (Programm für Deutschland 2017, Sections 8.8 and 8.9). Quran lessons are also rejected by "anti-constitutional mosque associations" (Programm für Deutschland 2017, Section 8.10).

In addition, religion in general or apart from Islam is only referred to twice in the federal party manifesto. Once in the section on "German Leitkultur" which is built on the values of Christianity, antiquity, humanism and the Enlightenment. Additionally, once in Section 10.1 where a subtle criticism of the Christian churches is evident upon their abolishment of the payment of church representatives, such as bishops, being taken from the general tax bill.

In comparison to the statements in the party manifesto of the "Republikaner" it is noticeable that the AfD focuses primarily on Islam as a culturally strange value system and tradition and as representing a politically dangerous power. Christianity, Judaism or other religions are not considered as religious factors; likewise, reference is only made once to the "Occident" as a cultural horizon. Rather, the party manifesto tries to avoid a Christian-religious undertone and is more religiously neutral or non-denominational. In its program for the European elections to the 9th European Parliament, currently the most recent election program, the AfD largely follows the lines of the federal election program of 2017. However, it is interesting that the AfD for the first time here refers to the danger of Islam to anti-Semitism, which must be consistently fought in word and deed. In addition to Islam, reference is only made once to Christianity and the churches are explicitly named; in the context of development aid, they should provide humanitarian aid (Europawahlprogramm 2019, p. 21).

## 5. Conclusions

The German AfD is less focused on religion than the three Dutch political parties, although their party manifestos have particular sections on religion, explicitly on Islam. A certain reluctance to make religious statements, also with regard to an evaluation of Christianity, can be seen clearly if one compares the AfD party manifestos to those of the

"Republikaner". With the "Republikaner", a Christian dominant culture is clearly being played off against an Islamic threat. In the AfD party manifestos, Islam is referred to as a religion, but in terms of content, religious characteristics and influences play no role. The lack of any reference to Jewish life in Germany, Israel or, with one exception, the increasing anti-Semitism evident in Germany is also striking. Religion is limited to the descriptive. In fact, this finding fits with Brubaker's thesis that populist movements in Europe emphasize Christianity as a cultural and civilizational identity (Brubaker 2017). The fact that the AfD has much support in the former Communist and anti-religious areas of Eastern Germany plays an important role here. If one wants to gain voters here, overly religious and Christian traditions should be avoided. Obviously, this also applies to AfD voters in Western Germany, where research has identified a high proportion of non-denominational voters (Hirscher 2020; Pfahl-Traughber 2017). In contrast to the "Republikaner" in the 1980s and 1990s, the success of the AfD voters shows not only the strategic absence of a religious word framing, but also a factual one: since the 1990s, the number of leavers from the two Christian churches in Germany has increased rapidly (Fowid 2020). This observation does not only apply to Christianity, but also to the notion of other religions than the monotheistic ones. In the election manifestos of both the German and Dutch political parties, Buddhism and Hinduism are completely absent. In comparison to the German AfD, the election manifestos of the three Dutch political parties contain a few more references to religion, but mainly in the service of taking a standpoint against Islam and/or to emphasize the Jewish-Christian-humanistic nature of the Netherlands. There are no specific chapters on religion. The manifestos include a few references to 'church' in the sense of the separation of or the relationship between faith community (church) and state, and the care for the historic-cultural heritage of church buildings. This finding also fits with Brubaker's thesis, whereby the Party Manifestos of the Dutch populist parties actually emphasize the "otherness" of a Christian-Jewish culture towards Islam more than the German ones (Brubaker 2017).

When looking at the first two decades of the 2000s, we see in the election manifestos different phases in the manifestation of populist political parties. In the manifestos of the LPF, the notion of religion is hardly present. The PVV especially is *the* anti-Islam political party. The keyword is 'Islam' and then, in the context of an anti-Islam attitude in view of mosques, headscarves and in general, the view that Islam is not a religion, but an ideology. The appearance of the keywords 'Israel' and 'Jerusalem' in manifestos (De agenda van hoop en optimisme 2010 (PVV), Hún Brussel, óns Nederland 2012 (PVV), Het gaat om u 2021 (PVV); Verkiezingsprogramma FvD 2017; Stem Nederland terug 2021 (FvD)) was unexpected.

The LPF, the PVV and the FvD present themselves in the political and societal domains as defenders of the Jewish-Christian-humanistic nature of the Netherlands, but the manifestos only include a few references to this nature, and sometimes, it is even absent (Politiek is Passie 2003 (LPF); Dit is niet het land wat ik voor mijn kinderen wil achterlaten 2006 (LPF); Hún Brussel, óns Nederland 2012 (PVV); Nederland weer van ons! 2017 (PVV); Verkiezingsprogramma FvD 2017; Stem Nederland terug 2021 (FvD)). For the FvD, this does not mean that religious freedom is unrestricted or limitless; it is subordinate to the freedom of expression as people (should) have the right to criticize religion and/or leave their religion. This correlates with the 'Religionskritik' in the election manifesto of the German AfD who, with reference to freedom of religion, conscience and expression, expressly supports the legitimacy of criticism of religion.

This research forms the starting point for further research. On the basis of our current research, we conclude that populist political parties are also about religion, although the notion of religion is virtually absent in their manifestos apart from an anti-Islamic standpoint and/or a few references to the Jewish-Christian-humanistic nature or culture of the Netherlands, as evident in the religious keywords search. Populist political parties frame religion for strategic, political and societal reasons. Nevertheless, in an upcoming article, we will present the results of further research on the question and answer of

whether the same religious keywords, as applied in this article, are relatively absent or more manifest in the election manifestos of several other Dutch and German political parties.

**Author Contributions:** Conceptualization, L.v.d.B. and K.K.; methodology, L.v.d.B. and K.K.; validation, L.v.d.B. and K.K.; formal analysis, L.v.d.B. and K.K.; investigation, L.v.d.B. and K.K.; resources, L.v.d.B. and K.K.; data curation, L.v.d.B. and K.K.; writing—original draft preparation, L.v.d.B. and K.K.; writing—review and editing, L.v.d.B. and K.K.; visualization, L.v.d.B. and K.K. All authors have read and agreed to the published version of the manuscript.

**Funding:** This research received no external funding.

**Conflicts of Interest:** The authors declare no conflict of interest.

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
