# Peer review of "Religion, Populism and Politics: The Notion of Religion in Election Manifestos of Populist and Nationalist Parties in Germany and The Netherlands"

_religions, doi:10.3390/rel12030178_

Round 1

Reviewer 1 Report

This article analyzes the manifestos of right-wing populist political parties in Germany and the Netherlands to understand how these parties invoke religion. The article shows that right-wing populist political parties in these countries generally condemn Islam and praise Judeo-Christian values, although religion is not the main theme of these manifestos.

Overall, I enjoyed this article and believe it will be of interest to scholars interested in issues related to Christian nationalism, the separation of church and state, religious freedom, and the intersection of religion and politics. I was particularly intrigued that political parties in these countries praise Christianity and condemn Islam in part because of Christianity's more favorable stance toward LGBTQ rights. The situation in countries like the United States is much different, as right-wing politicians there often invoke Christianity to attack LGBTQ rights. 

My main suggestions are:

(1) Since this is an English language journal, use English language subheadings (or at least provide English language translations of any non-English words within the subheadings themselves).

(2) Clean up the references. On the final page of the document, there appears to be at least one reference to a file stored on the author's personal computer, so a more formal citation to a file that would be accessible by readers is required.

(3) It would be helpful to include a summary table that lists all of the manifestos that are being analyzed, the countries and political parties associated with those manifestos, and the year of those manifestos' publication.

Author Response

Dear reviewer, 

Thank you for your kind words and helpful comments. In the revised article we implemented your comments:

  • we included a summary table with the title of the party manifestos, the year, the name of the political parties and the country
  • we replaced the Dutch titles of paragraphs by English translations
  • we tried to clear up the bilibography with view to the references to one of our computers. We do hope it is now better, as we did not quite understand the comment and/or could properly see what the problem might be. So, if it still not in good order, we are happy to change it again.

Kind regards, the authors

Reviewer 2 Report

This is a very good contribution to populism and religion literature. The author shows that for these parties, religion is a matter of identity politics but not even in the sense of belonging. It is about out-groups, otherisation and politics of fear. Highlighting this aspect and referring to the literature on religion and identity politics would greatly improve the paper. Moreover, even though this paper is on religion and populism, it only looks at this literature on tangentially. It would be great to engage this literature such as Roger Brubaker's work on civilisationism. Then in the conclusion, this paper's contribution to this literature can be explained.

Author Response

Dear reviewer, 

Thank you for your kind words and helpful comments. We read with great interest Brubaker and implemented quotes of and references to his work at several pages of our revised paper. 

We hope you will be satisfied. 

Kind regards, the authors